# Comparative effectiveness of adjunct non-pharmacological interventions on maternal and neonatal outcomes in gestational diabetes mellitus patients: A systematic review and network meta-analysis protocol of randomized controlled trials

**Sumanta Saha** [ORCID] *

Department of Community Medicine, R. G. Kar Medical College, Kolkata, West Bengal, India

* sumanta.saha@uq.net.au

## Abstract

### Background

Gestational diabetes mellitus (GDM) in pregnancy leads to a range of perinatal complications. Although several randomized controlled trials (RCT) have tested the effect of non-pharmacological standard GDM care adjuncts on these outcomes, there is no agglomerated statistical evidence on how their occurrence risk varies across interventions and with placebo. Therefore, a systematic review and network meta-analysis (NMA) protocol is proposed here to address this evidence gap.

### Materials and methods

A search for above RCTs published in the English language will transpire in PubMed, Embase, and Scopus databases irrespective of date and geographic boundary. The RCTs must test nutritional supplementation, digital intervention, structured exercise program, educational program, counseling service, or a combination of these prenatally in GDM patients. These should report $\geq 1$ of the following outcomes- cesarean section, pre-eclampsia, polyhydramnios, preterm birth, macrosomia, prolonged labor, gestational hypertension, premature rupture of membranes, congenital anomaly, Apgar scores, birth weight, birth length, gestational age at birth, neonatal hypoglycemia, neonatal hyperbilirubinemia, and neonatal Corpulence Index. The risk of bias assessment of the recruited trials will transpire using the Revised Cochrane risk-of-bias tool. Determination of the comparative effectiveness between interventions will occur by the frequentist method NMA for respective outcomes. The categorical and continuous outcomes effect size will get calculated in risk ratio and weighted or standardized mean difference, respectively. For each NMA model, network maps and league tables will show the connections between interventions and effect sizes with their 95% confidence intervals for each intervention pair compared, respectively. The publication bias assessment will occur using comparison-adjusted funnel plots. Best

**Funding:** The author received no specific funding for this work.

**Competing interests:** The author has declared that no competing interests exist.

intervention prediction for NMA models with statistically significant intervention effect will happen by determining the surface under the cumulative ranking curve values. Statistical analysis will ensue using Stata software (v16). The statistical significance estimation will happen at p<0.05 and 95% confidence interval.

## Trial registration

**PROSPERO registration no**: CRD42021271199; https://clinicaltrials.gov/.

## Introduction

Gestational diabetes mellitus (GDM) is glucose intolerance of any degree that occurs or gets detected for the first time during gestation [1]. Its prevalence among pregnant females depends on the criteria used to diagnose GDM [2]. The common perinatal complications of GDM include cesarean section, preeclampsia, macrosomia, neonatal hypoglycemia, new-born hyperbilirubinemia, and birth trauma [3]. Standard GDM care primarily includes medical nutrition therapy, lifestyle modification, and blood glucose monitoring. Insulin therapy is introduced when these conservative approaches are inadequate to achieve glycemic control. Research suggests that the benefits of standard GDM care are not uniform across all its perinatal outcomes. It decreases the risk of outcomes like preeclampsia, shoulder dystocia, and macrosomia [4]. On the other hand, the effects of standard GDM care didn't vary on certain outcomes like neonatal hypoglycemia and cesarean section from those not treated for GDM [4]. Therefore, it's crucial to investigate the role of adjunct prenatal non-pharmacological interventions (e.g., nutritional supplementation, digital intervention, exercise therapy, educational intervention) in GDM patients on various perinatal outcomes. Contemporarily several randomized controlled trials (RCTs) have researched it.

Findings from some of these trials are briefed here. Meta-analyses of trials studying the effect of antenatal vitamin D supplementation in GDM patients suggested a decreased risk of newborn hyperbilirubinemia, newborn hospitalization, macrosomia, and cesarean section [5–7]. A meta-analysis on omega-3 fatty acid supplementation in GDM patients depicted that the risk of preterm delivery, macrosomia, and 5 minute Apgar score, and newborn weight didn't vary from the placebo recipients [8]. A trial testing the role of face-to-face educational sessions in GDM patients found improvement of 1 and 5 minutes Apgar scores in the neonates [9]. However, it didn't observe any difference in the newborn birth weights and cesarean section (CS) requirements compared to standard GDM care recipients [9]. A prenatal dietary and exercise counseling testing trial in GDM patients reported a decreased birth weight and reduced risk of large for gestational age in newborns, but CS and preterm delivery risk didn't vary from the control group [10]. Another trial on GDM patients comparing twice-weekly exercise programs and standard GDM care didn't find the risk of CS, neonatal hypoglycemia, and newborn hyperbilirubinemia to vary [11]. Regarding a trial testing digital application in GDM patients, a smartphone-based app to inform about GDM didn't make any difference in the occurrence of chief maternal and neonatal outcomes [12].

Despite these trials, no standard guideline exists on prenatal non-pharmacological adjunct interventions in GDM patients. The plausible reasons are that these trials are underpowered (due to relatively small sample sizes) and/or nongeneralizable (as participant recruitment happened mainly from a particular nation) [5–7,9,11,12]. Moreover, meta-analysis studies (considered to be the highest level of epidemiological evidence) of RCTs in this context are unlikely

to be robust. For instance, summary estimates of meta-analysis of nutritional supplementation trials on GDM are based on very few clinical trials (≤5) [5–7]. Furthermore, these meta-analyses have tested a particular intervention (e.g., vitamin D supplementation); therefore, aggregated evidence across various nutritional supplements remains scanty. Meta-analyses are rare beyond nutritional supplementation type prenatal interventions (e.g., digital applications in GDM patients) in GDM patients. Altogether, there is a critical shortage of aggregated evidence comparing perinatal outcomes between non-pharmacological adjuncts themselves and with placebo for an overview of the existing evidence in the context of GDM patients.

Therefore, to address this evidence lacuna, a systematic review and network meta-analysis (NMA) protocol is proposed to study the congregated evidence based on existing clinical trials on how various adjuvant non-pharmacological antenatal interventions might affect the perinatal outcomes in GDM mothers. Due to the incorporation of NMA, this review will allow an indirect comparison between two interventions not compared in any real-life trial. The NMA models will compare the following interventions- nutritional supplementation, digital intervention, structured exercise program, structured educational program, counselling service, and a combination of these. Additionally, the outcomes of interest will get compared between respective nutritional supplements as its secondary aim. The list of perinatal outcomes of interest is available beneath the inclusion criteria below.

## Methods

A registered version of the proposed review is available at PROSPERO (Registration no: CRD42021271199) [13]. This report adheres to Preferred Reporting Items for Systematic Review and Meta-Analysis Protocols (PRISMA-P) (2015) reporting system (S1 File) [14].

### Eligibility criteria

#### Inclusion criteria.

1. Study population: GDM patients of any age irrespective of their history of GDM during previous gestations.

2. Study design: Placebo-controlled RCTs with any number of intervention arms irrespective of their follow-up duration.

3. Intervention arm: Along with standard/usual GDM care, trial participants in the intervention arm/s should receive any one or more of the following prenatal interventions:

   a. Nutritional supplementation (e.g., vitamin D and probiotics).

   b. Digital intervention/s (e.g., smartphone or smart-watch-based applications).

   c. Structured supervised exercise program.

   d. Structured educational and/or counseling service.

4. Comparator arm: Participants in the comparator arm should receive standard/usual GDM care with or without a placebo.

5. Outcomes of interest: The primary outcomes of interest will include the following perinatal outcomes:

   1. Cesarean section

   2. Pre-eclampsia

3. Polyhydramnios

4. Preterm birth

5. Macrosomia

6. Prolonged labor

7. Gestational hypertension

8. Premature rupture of membranes

9. Neonatal hypoglycemia

10. Neonatal hyperbilirubinemia

11. Congenital anomaly

12. Apgar scores at 1 min

13. Apgar scores at 5 min

14. Birth weight

15. Birth length

16. Gestational age at birth

17. Neonatal Corpulence Index

Outcome no. 12 onwards, the data must be reported in mean and standard deviation. A trial reporting at least one of these outcomes will be eligible for inclusion. Acceptance of the diagnosis of GDM and the definition of the maternal and neonatal outcomes and standard/usual GDM care will happen as per the trialists.

**Exclusion criteria.**

1. Non-RCT type study designs like observational studies and crossover studies.

2. Non-GDM type diabetes (e.g., type 1 or 2 diabetes).

## Information sources and search strategy

For eligible articles published in the English language, an electronic database search (in PubMed, Embase, and Scopus) will transpire irrespective of the publication date and geographic origin.

A search in the PubMed database will happen using the search strings below. The latter stemmed from key themes of the research question- GDM, clinical trial, and perinatal outcomes.

1. (("diabetes, gestational"[MeSH Terms] OR "gestational diabetes mellitus"[Title/Abstract] OR "GDM"[Title/Abstract]) AND "randomized controlled trial"[Publication Type]) AND (randomizedcontrolledtrial[Filter])

2. (("nutrition*"[All Fields] OR ("vitamin s"[All Fields] OR "vitamine"[All Fields] OR "vitamines"[All Fields] OR "vitamins"[Pharmacological Action] OR "vitamins"[MeSH Terms] OR "vitamins"[All Fields] OR "vitamin"[All Fields]) OR "probitic"[All Fields] OR ("synbiotics"[MeSH Terms] OR "synbiotics"[All Fields] OR "synbiotic"[All Fields]) OR ("prebiotically"[All Fields] OR "prebiotics"[MeSH Terms] OR "prebiotics"[All Fields] OR "prebiotic"[All Fields]) OR ("calcification, physiologic"[MeSH Terms] OR

("calcification"[All Fields] AND "physiologic"[All Fields]) OR "physiologic calcification"[All Fields] OR "mineralization"[All Fields] OR "mineral s"[All Fields] OR "mineralisable"[All Fields] OR "mineralisation"[All Fields] OR "mineralisations"[All Fields] OR "mineralise"[All Fields] OR "mineralised"[All Fields] OR "mineralising"[All Fields] OR "mineralizations"[All Fields] OR "mineralize"[All Fields] OR "mineralized"[All Fields] OR "mineralizer"[All Fields] OR "mineralizers"[All Fields] OR "mineralizes"[All Fields] OR "mineralizing"[All Fields] OR "minerals"[MeSH Terms] OR "minerals"[All Fields] OR "mineral"[All Fields]) OR ("digital"[All Fields] OR "digitalisation"[All Fields] OR "digitalised"[All Fields] OR "digitalization"[All Fields] OR "digitalize"[All Fields] OR "digitalized"[All Fields] OR "digitalizer"[All Fields] OR "digitalizing"[All Fields] OR "digitally"[All Fields] OR "digitals"[All Fields] OR "digitization"[All Fields] OR "digitizations"[All Fields] OR "digitize"[All Fields] OR "digitized"[All Fields] OR "digitizer"[All Fields] OR "digitizers"[All Fields] OR "digitizes"[All Fields] OR "digitizing"[All Fields]) OR ("smartphone"[MeSH Terms] OR "smartphone"[All Fields] OR "smartphones"[All Fields] OR "smartphone s"[All Fields]) OR ("exercise"[MeSH Terms] OR "exercise"[All Fields] OR "exercises"[All Fields] OR "exercise therapy"[MeSH Terms] OR ("exercise"[All Fields] AND "therapy"[All Fields]) OR "exercise therapy"[All Fields] OR "exercise s"[All Fields] OR "exercised"[All Fields] OR "exerciser"[All Fields] OR "exercisers"[All Fields] OR "exercising"[All Fields]) OR ("educability"[All Fields] OR "educable"[All Fields] OR "educates"[All Fields] OR "education"[MeSH Subheading] OR "education"[All Fields] OR "educational status"[MeSH Terms] OR ("educational"[All Fields] AND "status"[All Fields]) OR "educational status"[All Fields] OR "education"[MeSH Terms] OR "education s"[All Fields] OR "educational"[All Fields] OR "educative"[All Fields] OR "educator"[All Fields] OR "educator s"[All Fields] OR "educators"[All Fields] OR "teaching"[MeSH Terms] OR "teaching"[All Fields] OR "educate"[All Fields] OR "educated"[All Fields] OR "educating"[All Fields] OR "educations"[All Fields]) OR ("counsel"[All Fields] OR "counseled"[All Fields] OR "counselings"[All Fields] OR "counselled"[All Fields] OR "counselling"[All Fields] OR "counseling"[MeSH Terms] OR "counseling"[All Fields] OR "counsellings"[All Fields] OR "counsels"[All Fields])) AND (("diabetes, gestational"[MeSH Terms] OR "gestational diabetes mellitus"[Title/Abstract] OR "GDM"[Title/Abstract]) AND "randomized controlled trial"[Publication Type] AND "randomized controlled trial"[Publication Type])) AND (randomizedcontrolledtrial [Filter])

MeSH tagging of newly indexed articles in the PubMed database takes some time due to the manual nature of the process. Therefore, to identify such publications, free-text terms and phrases related to GDM were used with MeSH terms.

Relevant filters applied in the above search strings helped narrow the search results to the study design of interest.

These search strings were determined to be appropriate by using the following strategy. The respective search strings successfully retrieved three preidentified publications (testing exercise [11], nutritional [15], and digital [12] intervention in GDM patients) piloted for this purpose within the first 300 citations on relevancy-wise sorting of search results (S1 Table) [16]. An identical search strategy will apply to other databases.

Additional searches will ensue in the bibliography of articles recruited in this review and a preprint server, medRxiv [17].

## Study selection

The retrieved citations will get uploaded in the Rayyan systematic reviews software for study selection [18]. After eliminating the duplicate articles, two review authors will independently

skim through the titles and abstracts of the remaining and shortlist the seemingly eligible and dubious ones for full-text reading. The review authors will retain the list of excluded full-text-read articles.

## Data abstraction

After finalizing the articles to be reviewed, the review authors will independently abstract the subsequent data in pre-piloted forms [19,20] (S2 and S3 Files)-

1. Study details: Study design, the trial id, the nation/s where the trial got conducted, number of intervention arms, follow up duration of the trial, multi-centre or single centred trial, participant consent, ethical clearance, and funding information.

2. Study population details: The total number of participants randomized into each intervention arm and their mean age and sex distribution.

3. Intervention details: Interventions compared along with their administration frequency, duration, and dosages.

4. Outcome details: Perinatal post-intervention obstetric and neonatal outcomes of interest. When multiple outcomes' data are reported in an article, it will get captured in the S2 and S3 Files and subsequently reported in the summary table of the proposed review.

## Risk of bias (RoB) in individual studies

The Revised Cochrane risk-of-bias tool for randomized trials (RoB 2) will be used for the RoB assessment of the respective studies [21]. Utilizing signaling questions, the RoB will get assessed in the following bias domains- randomization process, intended interventions, missing outcome data, outcome data measurement, and reported results. Each of the signaling questions can have any of the following answers based on the review authors' judgment- yes, probably yes, probably no, no, and no information. For each domain, an assessment of the algorithm-based evaluation of the signaling questions will transpire to incorporate necessary changes. The overall RoB judgment will depend on the evaluations made for respective domains (detailed elsewhere) [21]. Two review authors will independently pursue the RoB assessment.

## Role of review authors

Three authors will be conducting the prospective review. The review authors will collate their findings after conducting each of the following independently- database search, study selection, data collection, and RoB assessment. By discoursing, they will try to resolve any conflict in an opinion. However, if this fails to bring resolution, a third-party opinion will be sought.

## Data synthesis

**NMA.** Frequentist method NMA will be performed. The NMA models will contrast the above-stated interventions of interest to determine their relative superiority for respective outcomes (NMA1). The interventions under respective parent intervention categories will get fitted in the NMA1 model as the latter. For instance, interventions falling under the nutritional supplements category like vitamin D, probiotics, and omega-3 fatty acids will get included in NMA1 as nutritional supplements instead of these discrete supplement types. Likewise, a structured exercise program in the NMA1 model will consist of all physical activity-related

interventions' (e.g., brisk walking, aerobics). The same rule will apply to the remaining interventions. Additional network meta-analyses models (NMA2) will compare how respective nutritional supplements (like vitamin D, probiotics, and omega-3 fatty acids) differ for each of the outcomes. Depending on the clinical type of outcome, a decrease or an increase in effect size (ES) will determine the effectiveness. For instance, a risk ratio of <1 will indicate a safer intervention for pre-eclampsia. In all NMA models, standard or usual GDM care recipients (with or without placebo) will form the common comparator. For categorical outcomes, NMA will transpire using the augmentation method, i.e., adding a small amount of data (0.5) to all intervention arms when zero events occur in an intervention arm [22].

**Criteria for selecting outcomes for NMA.** NMA will be performed for outcomes fulfilling the following conditions:

1. Low risk of heterogeneity:
   By conducting a pairwise meta-analysis (PMA), heterogeneity assessment will happen across trials testing the respective outcomes. The evaluation of the presence or absence of heterogeneity and its quantification will happen using $Chi^2$ statistics (statistical significance determined at p<0.1) [23] and $I^2$ values (of 25, 50, and 75% will categorize heterogeneity into low, moderate, and high categories, respectively) [24], respectively. This heterogeneity assessment will occur when at least 20 studies are available for PMA and/or the mean sample size is ≥80 for an adequately powered (80%) evaluation [25]. Since the interventions tested across the trials are unlikely to be identical for each outcome, random-effect PMA (inverse variance method) will be used for the heterogeneity assessment. For dichotomous outcomes with zero events in any intervention arm, 0.5 will be added to all cells of the 2x2 table.
   Outcome data of trials testing interventions in >1 arm will be combined for PMA. For continuous outcomes, the means and its standard deviation (SD) will be compared by following formulae (Eqs 1 and 2) [23].

$$\text{Mean} = \frac{(n1m1 + n2m2)}{n1 + n2} \tag{1}$$

$$\text{SD} = \sqrt{\left(\left((n1-1)sd1^2\right) + \left((n2-1)sd2^2\right) + \left(\frac{n1n2}{n1+n2}\right)\left(m1^2 + m2^2 - 2m1m2\right)\right)/((n1+n2)-1)} \tag{2}$$

where n1, n2, m1, m2, sd1 and sd2 are hypothetical sample sizes of intervention arm 1 and 2 of a clinical trial, mean value of arm 1 and 2, and standard deviation of m1 and m2, respectively.
   If the SD of the mean is unavailable for two groups of a trial, it will be calculated using the following steps:

   a. The t-value estimation will occur using the P-value of a t-test.

   b. Then, the SE calculation will transpire using the t value or confidence interval.

   c. Finally, the SD calculation will occur using the SE.
      The formulae for each of these steps are detailed elsewhere [23]. The calculated SD in this scenario will be assumed to be the same for both groups.
      PMA of outcomes with adequately powered heterogeneity assessment depicting $I^2 \leq 25\%$ and p-value of $Chi^2$ statistics <0.1 will get included in the NMA models.

2. The NMA model produces a connected network.

3. There is a degree of freedom for heterogeneity in a network to ensure a random-effect consistency model fitting.

4. There is a degree of freedom for inconsistency in a network to ensure an inconsistency model fitting.

**ES.** The ES estimation of categorical and continuous outcomes will ensue in the risk ratio and weighted or standardized mean difference, respectively, for both NMA and PMA.

**Network map.** The construction of network maps will transpire for a visual depiction of NMA models. It will help to understand the direct and indirect relationships among interventions. Each of the nodes and their thicknesses in the network map will denote a non-pharmacologic intervention and the number of participants receiving it, respectively. The line connecting two nodes will depict the trials that tested these interventions, and it will thicken with the number of trials testing it. Using an iterative method that swaps intervention pairs, excessive overlapping of lines caused complex network maps will be simplified [22].

**Transitivity and consistency.** A rationalized transitivity evaluation will ensue to identify potential effect modifiers. To ensure clinical homogeneity among participants across all trials, the proposed review's study population will not include patients with diabetes other than GDM.

Transitivity evaluation will also happen statistically by local (node-splitting method testing inconsistency in respective treatment pairs) and overall inconsistency assessment [26]. If both indicate an absence of inconsistency, a network consistency assumption will be accepted.

**League tables and ranking probabilities.** The calculated ES and its 95% confidence interval from respective NMA models will be presented in league tables. The diagonal cells across these tables will denote the contrasting interventions. When the same set of interventions is included in NMA models of two outcomes, for simplicity of presentation, their ESs will be placed in the same league table along with the upper and lower triangle of the league table (i.e., on either side of the diagonal cells).

NMA models of outcomes with at least one statistically significant efficacious finding will be subjected to best intervention prediction by successive estimation of the surface under the cumulative ranking curve values [27]. These values can range from 0–100%, where a higher value denotes a better-ranked intervention. Besides, cumulative ranking plots will be used for a visual presentation of estimated and predictive ranking [26].

**RoB across studies.** The assessment of the RoB across studies will include an evaluation of selective reporting by comparing the reported results with the pre-stated notion of the trialists. Additionally, a small study effect evaluation will ensue for each NMA model by constructing comparison-adjusted funnel plots. Asymmetric funnel plots will suggest the presence of variation in ESs between large and small studies.

**Sensitivity analysis.** A sensitivity analysis will repeat NMA1 and 2 by eliminating any trial with a high RoB component.

## Analytic tools

The 'meta' and 'network' packages of Stata statistical software version 16.0 (StataCorp, College Station, Texas, USA) will be used for PMA and NMA, respectively. The statistical significance estimation of all ESs will happen at $p < 0.05$ and 95% confidence interval.

## Reporting of the review

The proposed review report will follow PRISMA for Network Meta-Analyses statement guideline [28].

## Confidence in cumulative evidence

To assess the evidence quality of statistically significantly beneficial ESs in the NMA models, the Grading of Recommendations Assessment, Development and Evaluation (GRADE) approach proposed by GRADE Working Group (2004) will be used. This grading will label evidence into any of the following quality categories- high, moderate, low, and very low.

## Ethics and dissemination

As the proposed article will be a systematic review and network meta-analysis not requiring any direct human participation, an ethical clearance requirement will not apply. Once the review is complete, dissemination will happen by publishing it in an international journal and/ or conference presentation.

## Limitation

As review authors of the proposed review are adept in the English language only, articles published in any other language will not get included in this review.

## Supporting information

**S1 Table. PubMed search strategy.** Proposed search strategy in PubMed and its relevance. (DOCX)

**S1 File. PRISMA checklist.** Preferred Reporting Items for Systematic review and Meta-Analysis Protocols (PRISMA-P) 2015 checklist.
(DOC)

**S2 File. Proposed data abstraction form.**
(PDF)

**S3 File. Data abstraction form for analysis.**
(PDF)

## Author Contributions

**Conceptualization:** Sumanta Saha.

**Methodology:** Sumanta Saha.

**Visualization:** Sumanta Saha.

**Writing – original draft:** Sumanta Saha.

**Writing – review & editing:** Sumanta Saha.

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
