## [Decision Letter · Decision Letter 0]

6 Oct 2021

PONE-D-21-28908Comparative effectiveness of non-pharmacological interventions on maternal and neonatal outcomes in gestational diabetes mellitus patients: A systematic review and network meta-analysis protocol of randomized controlled trials.PLOS ONE

Dear Dr. Saha,

Thank you for submitting your manuscript to PLOS ONE. After careful consideration, we feel that it has merit but does not fully meet PLOS ONE’s publication criteria as it currently stands. Therefore, we invite you to submit a revised version of the manuscript that addresses the points raised during the review process. **Please also remember PLOS ONE does not copyedit accepted manuscripts, so the language in submitted articles must be clear, correct, and unambiguous.**

We look forward to receiving your revised manuscript.

Kind regards,

Jamie Matu, Ph.D.

Academic Editor

PLOS ONE

Journal Requirements:

3. We note that this manuscript is a systematic review or meta-analysis; our author guidelines therefore require that you use PRISMA guidance to help improve reporting quality of this type of study. Please upload copies of the completed PRISMA checklist as Supporting Information with a file name “PRISMA checklist”.

Reviewers' comments:

Reviewer's Responses to Questions

**Comments to the Author**

1. Does the manuscript provide a valid rationale for the proposed study, with clearly identified and justified research questions?

Reviewer #1: Yes

Reviewer #2: Partly

2. Is the protocol technically sound and planned in a manner that will lead to a meaningful outcome and allow testing the stated hypotheses?

Reviewer #1: Yes

Reviewer #2: Partly

3. Is the methodology feasible and described in sufficient detail to allow the work to be replicable?

Reviewer #1: No

Reviewer #2: Yes

4. Have the authors described where all data underlying the findings will be made available when the study is complete?

Reviewer #1: No

Reviewer #2: No

5. Is the manuscript presented in an intelligible fashion and written in standard English?

Reviewer #1: Yes

Reviewer #2: No

6. Review Comments to the Author

You may also provide optional suggestions and comments to authors that they might find helpful in planning their study.

Reviewer #1: In general this is a well designed systematic review / meta-analysis, that is already registered on PROSPERO. A good deal of thought has gone into the searches and terms used. I think it will make a useful contribution to the literature.

A couple of minor issues.

I would use the updated PRISMA guidelines and Checklist, rather than the 2015 version.

PMA isn't defined for the reader, presumably it is primary meta-analysis?

It isn't entirely clear what the effect sizes will be in the meta-analysis, and which formulas will be used to compute these effect sizes and so on.

Are the authors planning to look at other measures of bias, e.g. small study / publication bias? If so, what techniques will be used.

Reviewer #2: Major comments:

The use of non-pharmacologic lifestyle intervention (specifically, diet and exercise) and monitoring of blood glucose levels for the initial management of gestational diabetes (GDM) is standard of care. The determination of which of these accepted interventions has a more pronounced impact on pre-specific maternal and neonatal outcomes among women with GDM and their offspring would be of interest. However, the authors do not describe a study to compare effectiveness of currently accepted interventions for the management of GDM, but instead describe a comparison of non-pharmacologic interventions that are not adopted by current international guidelines due to a lack of evidence for benefit.

Thus, the hypothesis and goals of the review are not clear or consistent in the abstract or introduction of this manuscript. Furthermore, the validity of comparing across interventions such as omega-3 supplementation or smartphone applications, which prior studies did not show to differ from placebo, and why a network meta-analysis of these interventions should be performed, is questionable. Finally, the manuscript is not fully intelligible and some aspects are difficult to follow.

Minor comments:

-Inclusion criteria must be more specific: the types of interventions that will be included must be described and specified (eg., vitamin D, omega-3, low carb diet, etc. etc.); the specific outcomes that will be evaluated must be described and specified

-How many authors will perform the review specifically? (“3 or more”?)

-Page 10: “Both pharmacological and non-pharmacological interventions will not get included in the same NMA model.” – what pharmacologic interventions are referred to here, as the review purports to compare non-pharmacologic interventions?

7. PLOS authors have the option to publish the peer review history of their article (what does this mean?). If published, this will include your full peer review and any attached files.

Reviewer #1: No

Reviewer #2: No

---

## [Author Response · Author response to Decision Letter 0]

18 Oct 2021

RESPONSE TO REVIEWERS’ COMMENTS

Dear Reviewers,

 Thank you for reading the manuscript critically and sharing your insightful comments and suggestions. The manuscript has been thoroughly revised based on your advice. Additionally, several sentences have been edited to ensure a concise and logical presentation. Two manuscript copies, one with track changes enabled and another clean version with changes in the colored text, are also submitted. Below are the point-wise responses to your comments.

1. Does the manuscript provide a valid rationale for the proposed study, with clearly identified and justified research questions?

Reviewer #1: Yes

REPLY: 

Thank you for your feedback.

Reviewer #2: Partly

REPLY: 

Thank you for your comment. The rationale has been reinforced further based on your other comments below. A clarification has been added now to the ‘Introduction’ section (the last two paras particularly).

2. Is the protocol technically sound and planned in a manner that will lead to a meaningful outcome and allow testing the stated hypotheses?

Reviewer #1: Yes

REPLY: 

Thank you for your feedback.

Reviewer #2: Partly

REPLY: 

Thank you for your comment. The methodology has been elaborated further based on your comments below. Detailed responses are below (at ‘6. Review Comments to the Author’).

3. Is the methodology feasible and described in sufficient detail to allow the work to be replicable?

Reviewer #1: No

REPLY: 

Thank you for your comment. The analysis plan has been clarified further based on comments made by both reviewers. Please also see replies below about the amendments made in eligibility criteria and analysis (at ‘6. Review Comments to the Author’).

Reviewer #2: Yes

REPLY: 

Thank you for your feedback.

4. Have the authors described where all data underlying the findings will be made available when the study is complete?

Reviewer #1: No

REPLY: 

A data availability statement is now included in the manuscript just before the references.

Reviewer #2: No

REPLY: 

A data availability statement is now included in the manuscript just before the references.

5. Is the manuscript presented in an intelligible fashion and written in standard English?

Reviewer #1: Yes

REPLY: 

Thank you for your feedback.

Reviewer #2: No

REPLY: 

The manuscript has been hard edited meticulously. Besides responding to the reviewers' comments, every sentence was re-read and amended when required to ensure lexical integrity.

6. Review Comments to the Author

Reviewer #1: In general this is a well designed systematic review / meta-analysis, that is already registered on PROSPERO. A good deal of thought has gone into the searches and terms used. I think it will make a useful contribution to the literature.

REPLY: 

Thank you for finding the proposed review relevant.

A couple of minor issues.

I would use the updated PRISMA guidelines and Checklist, rather than the 2015 version.

REPLY: 

Thanks for the comment. By 'updated PRISMA guideline,' if you are referring to the PRISMA-2020 guideline for systematic review and meta-analysis, please allow me to clarify that it will not apply to this manuscript since this is a protocol for a systematic review and not the review itself. 

Nevertheless, based on your comment, I searched the PRISMA website for any updated version of the PRISMA-P statement. However, I couldn't find any updated version of PRISMA-P after 2015. Additionally, I contacted one of the authors of the PRISMA 2020 statement for the same, and I learned that the PRISMA-P 2015 version is the only version available for systematic review protocols presently. 

A humble request, if you are referring to any other version of the PRISMA statement, kindly let me know.

PMA isn't defined for the reader, presumably it is primary meta-analysis?

REPLY: 

I apologize for not mentioning the full form of the PMA. PMA stands for pairwise meta-analysis. The abbreviation's use has been updated accordingly in the manuscript.

It isn't entirely clear what the effect sizes will be in the meta-analysis, and which formulas will be used to compute these effect sizes and so on.

REPLY: 

Thank you for the comments. These issues are now addressed in the manuscript. I quote the sentence for your kind reference- 'The ES estimation of categorical and continuous outcomes will ensue in the risk ratio and weighted or standardized mean difference, respectively, for both NMA and PMA.' NMA stands for network meta-analysis.

Regarding formulae, these are specified in the updated manuscript in the ‘Methods’ section under the subheading ‘Criteria for selecting outcomes for NMA.’

Are the authors planning to look at other measures of bias, e.g. small study / publication bias? If so, what techniques will be used.

REPLY: 

Yes, publication bias assessment will happen. However, the description of it was incomplete in the manuscript. Thanks for pointing it out. Publication bias assessment will ensue using comparison-adjusted funnel plots, and this is now mentioned in the manuscript in the ‘Methods’ section (under subheading ‘RoB across studies’).

Reviewer #2: Major comments:

The use of non-pharmacologic lifestyle intervention (specifically, diet and exercise) and monitoring of blood glucose levels for the initial management of gestational diabetes (GDM) is standard of care. The determination of which of these accepted interventions has a more pronounced impact on pre-specific maternal and neonatal outcomes among women with GDM and their offspring would be of interest. However, the authors do not describe a study to compare effectiveness of currently accepted interventions for the management of GDM, but instead describe a comparison of non-pharmacologic interventions that are not adopted by current international guidelines due to a lack of evidence for benefit.

Thus, the hypothesis and goals of the review are not clear or consistent in the abstract or introduction of this manuscript. 

REPLY: 

Thank you for your comment. A modified rationale of the manuscript has been drafted (please see the last two paragraphs of the introduction section for these amendments). The manuscript now coins on the weaknesses of the existing trials and meta-analyses of these trials to reinforce its justification. Furthermore, to make the abstract and introduction consistent, the interventions and outcomes of interest are stated explicitly in both sections.

Furthermore, the validity of comparing across interventions such as omega-3 supplementation or smartphone applications, which prior studies did not show to differ from placebo, and why a network meta-analysis of these interventions should be performed, is questionable. Finally, the manuscript is not fully intelligible and some aspects are difficult to follow.

REPLY: 

Thanks for raising the issue. Your comment makes sense; a comparison between omega-3 supplementation versus smartphone application in terms of efficacy is perhaps two distant entities to make a meaningful comparison. Based on your comment, the analysis plan got updated to a more rationalized form. To describe using the above example, instead of comparing omega-3 supplementation versus smartphone, now juxtaposition will happen between the parent categories of these interventions (i.e., nutritional supplementation versus digital application).

The updated analysis plan will compare the following interventions- nutritional supplementation, digital intervention, structured exercise program, educational program, counseling service, or a combination of these. 

An additional network meta-analysis will separately compare various nutritional supplements to address your concern in the subsequent comment. It will help understanding how supplements like vitamin D, probiotics, omega-3 fatty acids, etc., differ in terms of the occurrence of the perinatal outcomes in GDM patients and their neonates.

Please find this updated plan introduced briefly in the 'Introduction' section and described elaborately under the 'Methods' section (beneath the 'NMA' sub-heading).

Minor comments:

-Inclusion criteria must be more specific: the types of interventions that will be included must be described and specified (eg., vitamin D, omega-3, low carb diet, etc. etc.); the specific outcomes that will be evaluated must be described and specified

REPLY: 

Thank you for the advice. The inclusion criteria have got updated now.

Regarding interventions', a list of interventions of interest has been specified (as stated in response to your previous comment). Also, following your suggestion, particular nutritional interventions will get compared in the additional network meta-analysis models (as specified in response to your previous comment). 

However, an explicit list of nutritional interventions (like vitamin D, probiotics, omega-3 fatty acids, etc.) to be studied in the additional network meta-analysis models isn't prepared to keep the scope of the review broad. It will plausibly enhance the comprehensiveness (by allowing all possible nutritional supplement types tested in clinical trials on GDM patients to get included in the review) and rigor of the evidence.

Next, the reason for not including interventions like a low carbohydrate diet and self-monitoring of blood glucose in the proposed review is that these are part of standard GDM care. Notably, the non-pharmacological adjuncts to standard GDM care are the interventions of interest. In other words, the prospective review aims to study non-pharmacological interventions' roles on perinatal outcomes when combined with standard/usual GDM care.

Concerning outcomes, now specific outcomes of interest have been enlisted under the inclusion criteria as per your suggestion. 

-How many authors will perform the review specifically? (“3 or more”?)

REPLY: 

Three authors will perform the review.

-Page 10: “Both pharmacological and non-pharmacological interventions will not get included in the same NMA model.” – what pharmacologic interventions are referred to here, as the review purports to compare non-pharmacologic interventions?

REPLY: 

Thanks for identifying the typing error. The sentence has been removed from the revised manuscript.

Thank you.

---

## [Decision Letter · Decision Letter 1]

4 Jan 2022

PONE-D-21-28908R1Comparative effectiveness of adjunct non-pharmacological interventions on maternal and neonatal outcomes in gestational diabetes mellitus patients: A systematic review and network meta-analysis protocol of randomized controlled trials.

PLOS ONE

Dear Dr. Saha,

Thank you for submitting your manuscript to PLOS ONE. After careful consideration, we feel that it has merit but does not fully meet PLOS ONE’s publication criteria as it currently stands. Therefore, we invite you to submit a revised version of the manuscript that addresses the points raised during the review process.

We look forward to receiving your revised manuscript.

Kind regards,

Maria G Grammatikopoulou

Academic Editor

PLOS ONE

Journal Requirements:

Additional Editor Comments:

Please implement the improvements suggested by the 2 authors.

Season's greetings!

Reviewers' comments:

Reviewer's Responses to Questions

**Comments to the Author**

1. Does the manuscript provide a valid rationale for the proposed study, with clearly identified and justified research questions?

Reviewer #3: Yes

Reviewer #4: Yes

2. Is the protocol technically sound and planned in a manner that will lead to a meaningful outcome and allow testing the stated hypotheses?

Reviewer #3: Yes

Reviewer #4: Yes

3. Is the methodology feasible and described in sufficient detail to allow the work to be replicable?

Reviewer #3: No

Reviewer #4: Yes

4. Have the authors described where all data underlying the findings will be made available when the study is complete?

Reviewer #3: No

Reviewer #4: Yes

5. Is the manuscript presented in an intelligible fashion and written in standard English?

Reviewer #3: Yes

Reviewer #4: Yes

6. Review Comments to the Author

You may also provide optional suggestions and comments to authors that they might find helpful in planning their study.

Reviewer #3: The authors provided us with a protocol for a systematic review and network meta-analysis for non-pharmacological interventions in gestational diabetes mellitus. The protocol is well structured and clear to read. Nevertheless, a variety of issues need to be covered before considered for publication.

1. Outcomes: It would be preferable to create an exhaustive list of outcomes based on the literature to ensure maximum objectiveness in the conducting of the review. This is not provided by the authors. Few examples are given e.g. preeclampsia. Still, it is to our belief necessary to prespecify (in the best possible way) outcome measures to be studied.

2. Details on reports providing information on more than one outcome have not been given.

3. Study selection: very weakly presented part. There is no specification of the program to be used or of the strategy (two separate authors etc). This needs to be analysed.

4. RoB evaluation using the 2011 tool needs to be justified. To our knowledge the tool has already been updated. Why is the newest form not chosen?

5. Prespecified forms of data extraction are not given. They are only described. Please provide a pdf version of the form.

6. Issues of the network meta-analysis are correct but are not backed up with bibliography. For example handling of zero events.

7. The biggest issue of the protocol is the search strategy. Even though information on the supplementary material is used to justify the choice of the protocol, we strongly disagree with the assumption of an RCT without adequate expansion of the search strategy. An in depth explanation is in order.

Reviewer #4: The present protocol is well designed and presented.

My comments are the following.

COMMENT 1. Only manuscripts in english are considered eligible for inclusion, please include more languages.

COMMENT 2. Please extend the search strategy to sources of grey literature.

7. PLOS authors have the option to publish the peer review history of their article (what does this mean?). If published, this will include your full peer review and any attached files.

Reviewer #3: No

Reviewer #4: No

---

## [Author Response · Author response to Decision Letter 1]

12 Jan 2022

Dear Reviewers,

I sincerely thank you for reading the manuscript and sharing your insightful comments. Please find two copies of the manuscript attached- one with track changes and the other a clean version. 

The manuscript file accompanies four supporting documents- 

1. S1 Table (relevancy wise search strings)

2. S1 File (PRISMA-P checklist)

3. S2 File (data abstraction form)

4. S3 File (data abstraction form for analysis)

Please find the responses to your comments below.

Review Comments to the Author

Reviewer #3: The authors provided us with a protocol for a systematic review and network meta-analysis for non-pharmacological interventions in gestational diabetes mellitus. The protocol is well structured and clear to read. Nevertheless, a variety of issues need to be covered before considered for publication.

1. Outcomes: It would be preferable to create an exhaustive list of outcomes based on the literature to ensure maximum objectiveness in the conducting of the review. This is not provided by the authors. Few examples are given e.g. preeclampsia. Still, it is to our belief necessary to prespecify (in the best possible way) outcome measures to be studied.

Author’s reply: 

Thank you for the feedback. The updated version of this manuscript contains the following 17 outcomes as stated under the ‘Methods’ section in the manuscript – 

1. Cesarean section

2. Pre-eclampsia

3. Polyhydramnios

4. Preterm birth

5. Macrosomia

6. Prolonged labor

7. Gestational hypertension

8. Premature rupture of membranes

9. Neonatal hypoglycemia

10. Neonatal hyperbilirubinemia

11. Congenital anomaly

12. Apgar scores at 1 min

13. Apgar scores at 5 min

14. Birth weight

15. Birth length

16. Gestational age at birth 

17. Neonatal Corpulence Index

2. Details on reports providing information on more than one outcome have not been given.

Author’s reply: 

Thank you for the comment. Now the following statement has been included in the revised manuscript to address the issue- “When multiple outcomes' data are reported in an article, it will get captured in the S2 and S3 Files and subsequently reported in the summary table of the proposed review.”

3. Study selection: very weakly presented part. There is no specification of the program to be used or of the strategy (two separate authors etc). This needs to be analysed.

Author’s reply: 

Thank you for the comment. The study selection section has been revised, and the updated version is quoted here for your kind reference- “The retrieved citations will get uploaded in the Rayyan systematic reviews software for study selection.[18] After eliminating the duplicate articles, two review authors will independently skim through the titles and abstracts of the remaining and shortlist the seemingly eligible and dubious ones for full-text reading.” Kindly note, as the subheading “Role of review authors” states how a conflict in an opinion between the review authors will get resolved, it's not restated here.

4. RoB evaluation using the 2011 tool needs to be justified. To our knowledge the tool has already been updated. Why is the newest form not chosen?

Author’s reply: 

Thank you for the comment. The updated version of the Cochrane RoB tool will be used. It’s quoted here from the revised manuscript for your kind reference- “The Revised Cochrane risk-of-bias tool for randomized trials (RoB 2) will be used for the RoB assessment of the respective studies.[19] Utilizing signaling questions, the RoB will get assessed in the following bias domains- randomization process, intended interventions, missing outcome data, outcome data measurement, and reported results. Each of the signaling questions can have any of the following answers based on the review authors' judgment- yes, probably yes, probably no, no, and no information. For each domain, an assessment of the algorithm-based evaluation of the signaling questions will transpire to incorporate necessary changes. The overall RoB judgment will depend on the evaluations made for respective domains (detailed elsewhere).[19] Two review authors will independently pursue the RoB assessment.” Kindly note, as the subheading “Role of review authors” states how a conflict in an opinion between the review authors will get resolved, it's not restated here.

5. Prespecified forms of data extraction are not given. They are only described. Please provide a pdf version of the form.

Author’s reply: 

Thank you for the comment. Please find a draft data abstraction form attached to the revised manuscript. Besides, a draft data collection sheet for capturing data for meta-analysis is also included. Kindly refer to the attached S2 File and S3 File.

6. Issues of the network meta-analysis are correct but are not backed up with bibliography. For example handling of zero events.

Author’s reply: 

Thank you for the feedback. Following additional relevant citations to back up the network meta-analysis are now included in the revised manuscript-

1. White IR. Network Meta-analysis. Stata J Promot Commun Stat Stata [Internet]. 2015;15:951–85. Available from: http://journals.sagepub.com/doi/10.1177/1536867X1501500403

2. Salanti G, Ades AE, Ioannidis JPA. Graphical methods and numerical summaries for presenting results from multiple-treatment meta-analysis: an overview and tutorial. J Clin Epidemiol [Internet]. 2011;64:163–71. Available from: https://linkinghub.elsevier.com/retrieve/pii/S0895435610001691

3. Rouse B, Chaimani A, Li T. Network meta-analysis: an introduction for clinicians. Intern Emerg Med [Internet]. 2017;12:103–11. Available from: http://link.springer.com/10.1007/s11739-016-1583-7

7. The biggest issue of the protocol is the search strategy. Even though information on the supplementary material is used to justify the choice of the protocol, we strongly disagree with the assumption of an RCT without adequate expansion of the search strategy. An in depth explanation is in order.

Author’s reply: 

Thank you for the advice. An expanded search string is now included in the revised manuscript. For your kind reference, I am pasting it here from the revised manuscript- ‘(("nutrition*"[All Fields] OR ("vitamin s"[All Fields] OR "vitamine"[All Fields] OR "vitamines"[All Fields] OR "vitamins"[Pharmacological Action] OR "vitamins"[MeSH Terms] OR "vitamins"[All Fields] OR "vitamin"[All Fields]) OR "probitic"[All Fields] OR ("synbiotics"[MeSH Terms] OR "synbiotics"[All Fields] OR "synbiotic"[All Fields]) OR ("prebiotically"[All Fields] OR "prebiotics"[MeSH Terms] OR "prebiotics"[All Fields] OR "prebiotic"[All Fields]) OR ("calcification, physiologic"[MeSH Terms] OR ("calcification"[All Fields] AND "physiologic"[All Fields]) OR "physiologic calcification"[All Fields] OR "mineralization"[All Fields] OR "mineral s"[All Fields] OR "mineralisable"[All Fields] OR "mineralisation"[All Fields] OR "mineralisations"[All Fields] OR "mineralise"[All Fields] OR "mineralised"[All Fields] OR "mineralising"[All Fields] OR "mineralizations"[All Fields] OR "mineralize"[All Fields] OR "mineralized"[All Fields] OR "mineralizer"[All Fields] OR "mineralizers"[All Fields] OR "mineralizes"[All Fields] OR "mineralizing"[All Fields] OR "minerals"[MeSH Terms] OR "minerals"[All Fields] OR "mineral"[All Fields]) OR ("digital"[All Fields] OR "digitalisation"[All Fields] OR "digitalised"[All Fields] OR "digitalization"[All Fields] OR "digitalize"[All Fields] OR "digitalized"[All Fields] OR "digitalizer"[All Fields] OR "digitalizing"[All Fields] OR "digitally"[All Fields] OR "digitals"[All Fields] OR "digitization"[All Fields] OR "digitizations"[All Fields] OR "digitize"[All Fields] OR "digitized"[All Fields] OR "digitizer"[All Fields] OR "digitizers"[All Fields] OR "digitizes"[All Fields] OR "digitizing"[All Fields]) OR ("smartphone"[MeSH Terms] OR "smartphone"[All Fields] OR "smartphones"[All Fields] OR "smartphone s"[All Fields]) OR ("exercise"[MeSH Terms] OR "exercise"[All Fields] OR "exercises"[All Fields] OR "exercise therapy"[MeSH Terms] OR ("exercise"[All Fields] AND "therapy"[All Fields]) OR "exercise therapy"[All Fields] OR "exercise s"[All Fields] OR "exercised"[All Fields] OR "exerciser"[All Fields] OR "exercisers"[All Fields] OR "exercising"[All Fields]) OR ("educability"[All Fields] OR "educable"[All Fields] OR "educates"[All Fields] OR "education"[MeSH Subheading] OR "education"[All Fields] OR "educational status"[MeSH Terms] OR ("educational"[All Fields] AND "status"[All Fields]) OR "educational status"[All Fields] OR "education"[MeSH Terms] OR "education s"[All Fields] OR "educational"[All Fields] OR "educative"[All Fields] OR "educator"[All Fields] OR "educator s"[All Fields] OR "educators"[All Fields] OR "teaching"[MeSH Terms] OR "teaching"[All Fields] OR "educate"[All Fields] OR "educated"[All Fields] OR "educating"[All Fields] OR "educations"[All Fields]) OR ("counsel"[All Fields] OR "counseled"[All Fields] OR "counselings"[All Fields] OR "counselled"[All Fields] OR "counselling"[All Fields] OR "counseling"[MeSH Terms] OR "counseling"[All Fields] OR "counsellings"[All Fields] OR "counsels"[All Fields])) AND (("diabetes, gestational"[MeSH Terms] OR "gestational diabetes mellitus"[Title/Abstract] OR "GDM"[Title/Abstract]) AND "randomized controlled trial"[Publication Type] AND "randomized controlled trial"[Publication Type])) AND (randomizedcontrolledtrial[Filter]).’

Thank you.

Reviewer #4: The present protocol is well designed and presented.

My comments are the following.

COMMENT 1. Only manuscripts in english are considered eligible for inclusion, please include more languages.

Author’s reply: 

Thank you for your comment. The review authors are adept in the English language only. Moreover, no funding is available for hiring language translators. It is a potential limitation of the proposed review, now mentioned in the revised manuscript under the subheading ‘Limitation’ using the following text- “As review authors of the proposed review are adept in the English language only, articles published in any other language will not get included in this review.”

COMMENT 2. Please extend the search strategy to sources of grey literature.

Author’s reply: 

Thank you for the comment. The search strategy has now been extended to a pre-print server medRxiv to include non-peer-reviewed articles. It has been stated in the revised manuscript using the following sentence- “Additional searches will ensue in the bibliography of articles recruited in this review and a preprint server, medRxiv.[17]”

Thank you.

---

## [Editor Report · Decision Letter 2]

18 Jan 2022

Comparative effectiveness of adjunct non-pharmacological interventions on maternal and neonatal outcomes in gestational diabetes mellitus patients: A systematic review and network meta-analysis protocol of randomized controlled trials.

PONE-D-21-28908R2

Dear Dr. Saha,

We’re pleased to inform you that your manuscript has been judged scientifically suitable for publication and will be formally accepted for publication once it meets all outstanding technical requirements.

Kind regards,

Maria G Grammatikopoulou

Academic Editor

PLOS ONE
---

## [Editor Report · Acceptance letter]

19 Jan 2022

PONE-D-21-28908R2 

Comparative effectiveness of adjunct non-pharmacological interventions on maternal and neonatal outcomes in gestational diabetes mellitus patients: A systematic review and network meta-analysis protocol of randomized controlled trials. 

Dear Dr. Saha:

I'm pleased to inform you that your manuscript has been deemed suitable for publication in PLOS ONE. Congratulations! Your manuscript is now with our production department. 

Kind regards, 

on behalf of

Dr. Maria G Grammatikopoulou 

Academic Editor

PLOS ONE